# Caveolae Mechanotransduction at the Interface between Cytoskeleton and Extracellular Matrix

**DOI:** 10.3390/cells12060942

**Published:** 2023-03-20

**Authors:** Laura Sotodosos-Alonso, Marta Pulgarín-Alfaro, Miguel A. del Pozo

**Affiliations:** Mechanoadaptation and Caveolae Biology Laboratory, Novel Mechanisms of Atherosclerosis Program, Centro Nacional de Investigaciones Cardiovasculares Carlos III (CNIC), 28029 Madrid, Spain

**Keywords:** caveolae, Caveolin1 (Cav1), mechanotransduction, plasma membrane (PM), cytoskeleton, actin, extracellular matrix (ECM), dolines

## Abstract

The plasma membrane (PM) is subjected to multiple mechanical forces, and it must adapt and respond to them. PM invaginations named caveolae, with a specific protein and lipid composition, play a crucial role in this mechanosensing and mechanotransduction process. They respond to PM tension changes by flattening, contributing to the buffering of high-range increases in mechanical tension, while novel structures termed dolines, sharing Caveolin1 as the main component, gradually respond to low and medium forces. Caveolae are associated with different types of cytoskeletal filaments, which regulate membrane tension and also initiate multiple mechanotransduction pathways. Caveolar components sense the mechanical properties of the substrate and orchestrate responses that modify the extracellular matrix (ECM) according to these stimuli. They perform this function through both physical remodeling of ECM, where the actin cytoskeleton is a central player, and via the chemical alteration of the ECM composition by exosome deposition. Here, we review mechanotransduction regulation mediated by caveolae and caveolar components, focusing on how mechanical cues are transmitted through the cellular cytoskeleton and how caveolae respond and remodel the ECM.

## 1. Introduction

Cells and tissues are subjected to multiple forces that reach the plasma membrane (PM), which orchestrates signal transduction in response to the mechanical forces by acting as a scaffold because of its association with the intracellular cytoskeleton (CSK) and the extracellular matrix (ECM) [1,2,3]. In this process, specialized domains and the organization of the PM are essential. Caveolae are small PM invaginations identifiable by electron microscopy because of their Ω-shaped morphology and diameter of 50–80 nm. They are highly dynamic structures characterized by enrichment in cholesterol and sphingolipids. Caveolae are present in many cell types and tissues, but their density varies considerably. In general, they are more abundant in mechanically active tissues such as adipocytes, endothelial and muscle cells, and fibroblasts [4,5].

The cytoskeleton is also key in mediating mechanotransduction; it organizes the intracellular content of the cell and connects and generates forces between the cell and the surrounding microenvironment [6,7,8]. Interestingly, caveolae associates with the dynamic CSK, which is crucial for both the sensing and transducing of mechanical forces into the cell. 

ECM composition and organization are also relevant aspects in the global picture of mechanobiology, since they dictate ECM mechanical properties, thus determining some of the mechanical cues sensed by cells. However, reciprocity between ECM mechanosensing and remodeling by cells has been established. Integrins have been extensively studied as bidirectional links between the ECM and the CSK (reviewed in [9]). However, the aim of this review is to highlight the role of the caveolae and caveolar components, mainly Caveolin1 (Cav1), on the integration of mechanical forces transmitted through the ECM and the specific role of the modulation of the cellular cytoskeleton as an effector mechanism by which caveolae and caveolins are able to respond and modify ECM [10,11,12].

## 2. Caveolae: Composition, Organization, and Function

### 2.1. Caveolae Composition and Organization

The main components of caveolae are two families of proteins: caveolins and cavins. There are three caveolin proteins in mammals: Caveolin1 (Cav1), Caveolin2 (Cav2), and Caveolin3 (Cav3). Cav1 and Cav2 are expressed in most tissues, whereas Cav3 is mainly expressed in muscle cells [13]. The depletion of Cav1 and Cav3 leads to the complete absence of caveolae in tissues where they are normally expressed [14,15,16], while Cav2 deletion does not affect caveolae formation [17]. 

Caveolins are small integral membrane proteins with the N- and C-termini facing the cytoplasm, so they adopt a hairpin conformation. Cav1 can be translationally modified in several residues. At the C-terminus, it could be palmitoylated, which could modulate its interaction with the PM [18,19,20]. At the N-terminus, Cav1 could be ubiquitinated, which regulates its sorting into multivesicular endosomes and its degradation [21,22,23]. Moreover, Cav1 can be phosphorylated on a conserved tyrosine-14. This phosphorylation is mediated by the Src kinase [24], but also through other kinases such as Fyn or Abl [25,26]. This modification affects caveolar endocytosis [27,28,29], Cav1 oligomerization, and interaction with other caveolar proteins and lipids [24,30,31,32]. Interestingly, this phosphorylation also occurs in response to mechanical stress, such as cyclic stretching, and it promotes the expression of caveolar genes to increase caveolae numbers at the PM [30,33] and upon shear flow exposure [34]. It regulates RhoA activation, actin dynamics, contractility, and cell migration [10,35], and modulates focal adhesion dynamics [36] and ECM remodeling [11]. The regulation and implications of Tyr-14 Cav1 phosphorylation (pY14Cav1) have recently been reviewed [37,38]. Cav1 can also be phosphorylated on serine-80 by PKC, which regulates its PM insertion and cholesterol trafficking [39]. 

The other major components are cytoplasmic cavin proteins. This family is com-posed of four different proteins: Cavin1 (also known as PTRF, polymerase I transcription release factor), Cavin2 (SDPR, serum-deprivation response protein), Cavin3 (SRBC/PRKCDBP, Sdr-related gene product that binds to c-kinase) and Cavin4 (MURC, muscle-restricted coiled-coil protein), which is expressed in muscle tissues [40]. They are peripheral membrane proteins that form oligomeric complexes with caveolins in regions of the PM to form caveolae [41,42]. Cavin1 is essential for the correct formation of caveolae [43,44], whereas Cavin2, Cavin3, and Cavin4 have regulatory functions and contribute to caveolae stabilization and functionality [45,46,47]. In the absence of Cavin1, caveolae are not formed, but Cav1 is able to engage in PM invaginations of different sizes, usually bigger than caveolae. The recently discovered ‘dolines’ sense low- to medium-range forces, which are more predominant in physiological contexts, thus conferring regular mechanoadaption and mechanoprotection to tissues devoid of caveolae (Figure 1) [48]. In fact, caveolae are more abundant in tissues subjected to high mechanical forces, such as lungs, skeletal muscle, the heart, vessels, and white adipose fat [4,13]. Dolines coexist with caveolae, but whether they are present in caveolae-free cells such as neurons or lymphocytes is a subject of current research.

Furthermore, other accessory proteins are recruited to the neck of caveolae, which are Pacsin2 (also called Syndapin2, PKC, and casein kinase substrate in neurons 2) and EHD2 (Eps-15 homology domain-containing protein 2), a dynamin-like ATPase that controls caveolae dynamics and caveolar neck morphology. Pacsin2 is an F-BAR (Bin-amphiphysin-Rvs) protein that regulates the characteristic membrane curvature of caveolae [49] and it also binds Dynamin-2, which contributes to caveolar scission [50]. Pacsin2 and Pacsin3 regulate caveolar density [51] and Pacsin2 could be phosphorylated by protein kinase C (PKC) and contribute to caveolar disassembly [52]. EHD2 is a dynamin-like ATPase that controls the dynamics and neck morphology of caveolae [53,54,55,56]. The deletion of this protein promotes increased caveolar mobility [57]. It has been reported that other EHD proteins can be recruited to caveolae, such as EHD1 and EHD4, which are present in caveolae when cells lack EHD2 [58]. When the three EHD proteins are absent, caveolae cannot cluster and are more susceptible to rupture upon mechanical forces [58]. Moreover, the transmembrane receptor ROR1 (receptor tyrosine kinase-like orphan receptor 1) is also present in caveolae, but it seems to only be relevant during embryonic and fetal development and contributes to maintaining the interaction between caveolins and cavins. ROR1 is unlikely to be a universal regulator of caveolae formation [42,59,60]. Finally, the F-BAR protein FBP17 is important for the formation of high-order caveolar structures or rosettes [61] (Figure 2).

In summary, as an oversimplification, it has been estimated that a single and mature caveola contains ~150 Cav1 molecules, ~29 Cav2 molecules, ~50–80 Cavin1 molecules, ~20 Cavin2,3 molecules, and ~40 EHD2 molecules [42,53,62,63,64,65]. 

As mentioned previously, caveolae are PM domains enriched in certain lipids; there-fore, they also play an important role in the correct formation of caveolae. Cav1 binds to cholesterol [66,67] and can also be present in lipid droplets [68]. In the absence of Cav1, cholesterol tends to accumulate in intracellular compartments and affects proper cell functioning, as it alters the endoplasmic reticulum, mitochondria, and the membranes associated with both [67,69,70]. Lipids play a crucial role in caveolae formation and stabilization and the recruitment of most caveolar components [13,71]. Alterations in cholesterol content also significantly affect Cav1 and caveolae, as cholesterol depletion by methyl-β-cyclodextrin treatment leads to the dissociation of cavins, and the caveolae flatten [5,72]. Moreover, cavins bind to PtdIns(4,5)P2 (phosphatidylinositol 4,5-biphosphate) and phosphatidylserine (PtdSer) [43,66]. EHD2 can bind to PtdIns(4,5)P2 [73] and requires cholesterol to bind to caveolae [54], and neck morphology is subjected to changes in lipid accumulation [71]. Pacsin2 and FBP17 can also bind to phosphatidylinositol [74]. These phospholipids are important for caveolae stability and formation in the PM [75]. 

Caveolae are highly dynamic and can adopt different levels of organization. Caveolae can be organized at the PM as single pits or simple caveolae, but they can also form higher-order structures called rosettes or clusters of caveolae, characterized by multiple caveolae that connect through a single neck to the PM [4,5,58]. These structures are formed when cell tension is reduced, as occurs upon loss of cell adhesion [76,77], and they can flatten faster than single pits in response to mechanical challenges [61]. These structures require FBP17 [61], but EHD proteins also regulate their formation [58] (Figure 2). Recently, dolines have been described as other PM invaginations regulated by Cav1 that can also sense mechanical forces and have a different morphology than caveolae and rosettes (Figure 1) [48]. All of these caveolar- and Cav1-related structures allow the PM to respond differently according to the stimuli perceived.

### 2.2. Caveolae as Mechanosensors and Mechanotransducers

Caveolae are involved in diverse functions, such as lipid and cholesterol homeostasis [78,79,80,81], cell adhesion, polarization and migration [10], cell cycle regulation [82], and anchorage-dependent growth [83].

As cholesterol-enriched membrane microdomains (CEMMs), they participate in cellular signaling [84] by acting as platforms that sequester and compartmentalize proteins and molecules at the PM, such as transforming growth factor β receptors (see below) [85,86,87], insulin receptors [88], and epidermal growth factor receptors [89]. Caveolae participate in clathrin-independent endocytosis of certain membrane receptors [90,91]. Apart from membrane receptors, caveolae were proposed to participate in the endocytosis of other cargoes, such as SV40 virus or the beta subunit of cholera toxin (ChTxB) [92]. However, it is still a controversial topic, given that caveolae-independent endocytic pathways have been described for these cargoes [93], and no caveolae-specific cargoes have been identified until now. There is evidence that SV40, ChTxB, and others enter the cell through caveolae, but they can enter equally well in its absence. We recently proposed that the mechanosensing properties of caveolae are coupled to their membrane-trafficking abilities, in that caveolae can provide membrane where it is needed as a means to buffer PM tension increases [13]. 

Pertaining to the scope of this specific issue, caveolae also participate in mechanosensing and mechanotransduction [4,42]. In this context, caveolae are able to flatten in response to increased PM tension, as it occurs by cell stretching or osmotic swelling [94] (Figure 2). It has also been observed that stretching significantly reduces caveolar clusters or rosettes [95]. This acts as a protective response that buffers increased membrane tension, preventing PM rupture, and activating downstream pathways, such as MAP and SRC kinases, RHO, and RAC small GTPases [4,94]. Cells and tissues depleted of caveolae are more prone to increased PM damage in comparison to wild-type ones when subjected to mechanical stress. Cav1 and other caveolar proteins, as EHDs, contribute to maintaining proper PM integrity under stretching. In the absence of EHD proteins, mechanical stretching results in a reduction in caveolae and the cells are more prone to PM rupture [58]. Upon cyclic stretching, the phosphorylation on Cav1 Y14 regulates the transcriptional regulation of Cav1 and Cavin1 to contribute to caveolae formation [30,96,97]. In this way, Cav1 phosphorylation seems to also be involved in cell mechanical protection. 

Caveolae are then considered membrane reservoirs available to rapidly respond to a mechanical challenge; this assumption is supported by the fact that caveolae cover up a high proportion of the total PM area in cells subjected to mechanical challenge [98].

During caveolae flattening, some caveolar components are released and become available to perform non-caveolar-dependent functions, a key process for their function as mechanotransducers. For example, cavin proteins are released from the PM when caveolae are disassembled. Cavin1 has been localized in the cell nucleus, where it regulates rRNA transcription in response to insulin stimulation and osmotic stress [99,100]. Cavin3 could also enter the nucleus when caveolae are disassembled by ultraviolet light exposure, and nuclear Cavin3 regulates DNA damage and apoptosis through the interaction with the phosphatase PP1α [101] and BRCA1 [102]. Finally, upon mechanical stretching and osmotic shock, EHD2 could also translocate into the cell nucleus, where it represses caveolar gene transcription through KLF7 [103] (Figure 2).

Another mechanical stimulus that has been reported to be sensed by caveolae is shear stress. In 1999, Rizzo et al. identified a high density of caveolae in the luminal surface (which is exposed to hemodynamic forces) of rat lung endothelial cells [104]. They found eNOS localized at the caveolae, where this enzyme is inhibited by interactions with Cav1, but activates in response to flow in a Cav1-mediated mechanism. Then, Cav1 could regulate vascular tone by controlling the production of nitric oxide [104]. Moreover, exposure to shear modulates the expression of Cav1 and the formation and localization of caveolae at the apical membrane of endothelial cells [105,106]. The phosphorylation of Cav1 at Tyr14 in response to flow [34] might underlay, at least in part, the increase in Cav1 expression upon shear [30]. 

The response of endothelial cells to shear depends on the velocity, direction, and frequency of the flow (i.e., pulsatile or not). Unidirectional slow shear (<12 dyn/cm^2^) or bidirectional/disturbed flow, also known as oscillatory shear stress (OSS), results in endothelial cell inflammation, while undisturbed laminar flow (with values of >12 dyn/cm^2^), or laminar shear stress (LSS), is protective [107]. In fact, atherosclerosis is preferentially developed at curvatures and branching points of the vascular tree, which are subjected to OSS [108]. The role of Cav1 in vascular mechanotransduction and remodeling has been described [109,110]. In addition, protection from atherosclerosis development in Cav1-KO mice was reported [110,111], emphasizing the ability of Cav1-expressing cells to discriminate between LSS and OSS. However, independently of mechanosensing, caveolae have been reported to be involved in other processes that might be relevant for atherosclerosis development, such as LDL transcytosis [112], inflammation or cholesterol efflux [113]. 

Cav1 can also modulate YAP (yes-associated protein) and TAZ (transcriptional coactivator with PDZ-binding motif) [12,114,115], which is regulated by mechanical forces (see below), in turn modulating the expression of genes involved in proliferation, migration, and differentiation [116,117,118]. 

Additionally, a commonly underestimated function of caveolins is cell–ECM cross-talk. The sensing of the mechanical properties of ECM by caveolar components occurs mainly at the focal adhesion (FA) level, macromolecular assemblies that occur at cell–ECM contact sites, where the contractile machinery of the cell (i.e., actin filaments) and integrins connect through adaptor proteins. Cav1 is involved in integrin-dependent signaling [34,119] and in FA assembly, maturation, and turnover [10,120]. ECM remodeling and deposition are key processes by which cells respond to substrate stiffness and organization in order to remodel it accordingly. Caveolae/Cav1 implications on these processes will be discussed more deeply in this review.

### 2.3. Caveolae in Disease

Mutations in caveolar components have been linked to several pathologies called ‘caveolinopathies’. As mentioned, caveolae are more abundant in adipocytes, endothelial cells, and muscle cells; therefore, caveolae gene mutations are often associated with lipo-dystrophy, pulmonary arterial hypertension (PAH), muscular dystrophies, cardiomyopathies, and cancer [5,31]. Mice lacking Cav1 present a lipodystrophic phenotype, PAH, and cardiac disease [14,16,121,122], whereas Cav3 knockout mice, as expected from the muscle expression of Cav3, present cardiomyopathies and muscle disorders [123]. Human mutations in Cav1 have been described that cause lipodystrophy [124,125,126] and also in Cavin1 that lead to a similar phenotype [127,128]. Moreover, mutations in Cav1 have been associated with PAH [129,130,131,132,133]. Regarding Cav3, mutations in this gene have also been reported that lead to muscle diseases such as limb-girdle muscular dystrophy, rippling muscle disease, hyperCKemia, distal myopathy, and hypertrophic cardiomyopathy [134,135]. As Cavin1 is expressed in muscle and non-muscle tissues, Cavin1 mutations in humans cause lipodystrophies associated with myopathy, long-QT syndrome, and fatal cardiac arrhythmias [127,128,136], and Cavin4 mutations are linked to cardiac disease [137]. However, the molecular mechanisms that fully explain how these different mutant proteins cause this broad phenotype are not completely understood, although they could be partially explained by an impaired mechanoprotection response. As mentioned, Cav1 depletion is associated with atherosclerosis protection [110,111,112,113], although the responsible molecular mechanism(s) has not been elucidated. The role of Cav1 in cancer has also been widely studied, although the specific involvement of this protein as a tumor suppressor or oncogene seems to be context- and cell-dependent [138,139]. Finally, antifibrotic properties have been associated with Cav1 [140,141], being involved in a variety of fibrotic diseases such as systemic sclerosis or pulmonary fibrosis [142,143]. Interestingly, Cav1 in stromal cells has been also reported to promote tumor invasion and metastasis through the modification of the ECM [11,139].

## 3. Interaction between Caveolae and Cytoskeleton

The cytoskeleton is crucial in the response to mechanical insults. The CSK is formed by different proteins that can assemble into larger structures to form different types of filaments and networks, with diverse organization, functions, and properties. There are three types of cytoskeletal components and caveolae that have been associated with each of them: actin filaments [95], microtubules [144], and certain types of intermediate filaments [145]. Here, we will describe how these associations occur between each cytoskeletal component and caveolae and how they reciprocally regulate its organization, dynamics, and function.

### 3.1. Actin Cytoskeleton and Caveolae

Caveolae are in close proximity to the cortical actin beneath the PM, specifically the stress fibers. This association relies on the interaction between different caveolar components and actin filaments both directly and indirectly. The first evidence was obtained by electron microscopy images [146,147,148,149]. Using a 3D high-resolution reconstruction of electron microscope tomography images, the existence of complex connections between caveolae, actin, and microtubule filaments has been demonstrated [150]. In addition, there is a co-alignment between these actin stress fibers with Cav1 in many different cell types [95,151]. This association is partially caused by the interaction between the previously mentioned caveolar components with actin filaments and actin-regulating proteins through protein linkers. Proteomics of caveolar-enriched PM fractions have revealed that approximately 33% of the proteins identified are cytoskeletal proteins or involved in cytoskeletal regulation, such as actin beta and gamma, annexin A2 and V, myosin, tubulin, and vimentin, among others [152]. 

Cav1 could directly interact with Filamin-A, an actin cross-linker protein present in the stress fibers [153,154], and with Tropomyosin-3.1, which tightly associates with actin, and also binds Cavin1 [151]. The caveolar neck proteins are also linked to the actin filaments, as EHD2 that associates with F-actin [54,55] and Pacsin2 that binds directly to F-actin through its F-BAR domain [155] and through its SH3 domain to dynamin-2, N-WASP, and the small GTPase Rac1, which regulate actin cytoskeleton [52,156,157]. Cavin-2 colocalizes with cortactin, which recruits actin polymerization proteins, such as Arp2/3 [158]. Additionally, Cavin3 and ROR1 can also bind to the motor protein Myosin-1c, involved in cellular transport and endocytosis regulation, controlling caveolae presence at the PM [60,159,160]. 

The advent of novel advanced proteomics techniques, such as proximity-dependent biotin identification (BioID) analysis [161], allowed the identification of many potential interactors of the different caveolar components, and actin-related proteins are among the most relevant. Among the Cavin1 interactors in HeLa cells, there are several proteins related to actin filaments, such as CD2AP (CD2-associated protein), and microtubules, as MAP4 [162]. The intracellular interactome of Cavin3 has also revealed its ability to bind to actin-related proteins, such as Annexin A2, Cofilin, or ACTR3 (Actin-related protein 3) [101]. However, whether these interactions occur in the PM has not yet been demonstrated. In vivo, the Cavin4 and Cavin1 interactome in zebrafish skeletal muscle has been identified, of which PM proteins were the majority, but there were also many related to actin filaments, such as actin, myosin, dystrophin, ankyrin3a, spectrin-alpha, and limch1a, which is involved in stress fibers and focal adhesion assembly [163].

Cav3 is associated with different components of the dystrophin–glycoprotein com-plex (DGC), which interacts with cytoskeletal proteins [164,165]. T-tubule formation is dependent on Cav3, Cavin4, and actin-organizing proteins such as Bin1 and N-WASP [166,167,168,169,170]. Recently, approximately 10% of the proteins identified in the Cav3 interactome were cytoskeletal proteins, such as actin, myosin, and tropomyosin [171]. 

Collectively, these complex connections between caveolar components and actin filaments highlight the mechanistic interplay between them, as both can regulate each other in response to mechanical challenges.

#### 3.1.1. Regulation of Caveolae by the Actin Cytoskeleton

The actin cytoskeleton plays an important role in many mechanotransduction pathways and regulates caveolae organization in the PM by several pathways. Actin filaments regulate caveolae internalization by confining and organizing them at the PM, and negatively regulate the number of caveolar rosettes at the PM [172,173]. Caveolae organization is also regulated by the actin cytoskeleton regulators Abl and mDia1. Abl is a tyrosine kinase that phosphorylates actin regulators, and also Cav1 [25,174]; formin mDia, which is downstream of Abl, is involved in linear actin fiber polymerization [76,175]. In the absence of these proteins, stress fibers co-aligned with Cav1 are reduced and Cav1 clusters increase, which affects rosette formation upon cell detachment [76]. Abl and mDia maintain caveolae attached to the actin cytoskeleton and therefore regulate caveolar organization at the PM. Filamin-A also contributes to caveolae stabilization at the PM by acting as a linker between Cav1 and stress fibers [153]. Recently, the depletion of formin proteins FHOD1 and Dia1 has also been shown to decrease Cav1 vesicle movement, especially when cells are subjected to softer matrix stiffness or hypo-osmotic shock [176]. The disruption of stress fibers also affects Cav1 vesicle localization and reduces Y14Cav1 phosphorylation [151]. The formin-binding protein FBP17 regulates caveolar rosettes and is required for buffering PM tension upon hypoosmotic shock because when PM tension-increased c-Abl phosphorylates FBP17 and releases mDia1 from inhibition, this buffers the augmented tension [61]. Cavin1 and Cav2 PM localization are also regulated by actin filaments and microtubules [177,178]. 

All of this evidence reinforces the notion that the association between caveolar components and actin is crucial for the proper organization of caveolae at the PM, and consequently for its correct functioning.

#### 3.1.2. Converse Regulation of Actin Filaments by Caveolar Components 

Actin filaments are dynamic structures, and their assembly and disassembly are regulated by actin-binding proteins in response to different stimuli. As actin modulates caveolar organization, caveolar components also affect actin fiber rearrangement. Cav1 depletion reduces the number of stress fibers, whereas Cav1 overexpression increases the fluorescence intensity of actin and its reorganization [179,180]. Among the proteins that regulate actin polymerization, the Rho small GTPases RhoA, Rac1, and Cdc42 play an essential role in its remodeling and many processes regulated by them, such as cell migration, division, and polarity [181,182,183]. Caveolae also regulate Rho GTPases. The loss of Cav1 impairs actin cytoskeleton equilibrium by the increased phosphorylation of AMPK (AMP-activated protein kinase that regulates small Rho GTPases [184], and it reduces RhoA-myosin II activation and increases the activity of Rac1/Cdc42-Pak1-cofilin, which results in a decrease in thick actin stress fibers and an increase in lamellipodia [10,151,185,186,187] through the activation of p190RhoGAP [10,11,35,188]. Caveolar components can also regulate RhoA signaling through other mechanisms, such as the PM targeting of Rac1, physical interactions, and rates of degradation, among others, and therefore influence cytoskeletal regulation [10,28,95,189,190]. This will be further explained in the context of ECM remodeling (see below). Interestingly, non-caveolar Cav1 present at the apical membrane of the primary cilium could regulate ciliary length and rearrange the apical actin meshwork via RhoA and its effectors ROCK and Dia1 [191]. It has also been reported that the absence of Cav1 in epithelial monolayers promotes the recruitment of formin FMNL2 to the cell cortex and cell junctions to enhance contractile tension via a lipid-based signaling pathway [192]. This compromises cell morphology and directional cell migration [151,193]. A lack of Cav1 also promotes S-gutathionylated actin at the PM, which affects polyamine uptake [194]. Taken together, caveolae could modulate actin dynamics by controlling ac-tin-regulated proteins such as RhoGTPases and formins, and it is probable that, similar to non-caveolar Cav1, the other caveolar components could also regulate the actin cytoskeleton outside caveolae.

### 3.2. Microtubules and Caveolae

Microtubules are crucial for membrane trafficking, as they act as tracks for vesicle and organelle movements. They are also involved in caveolae trafficking and recycling. In contrast to the actin cytoskeleton, Cav1 internalization is inhibited by the depletion of microtubules upon treatment with nocodazole, and it also leads to an increase in Cav1 linear organization at the PM and in the number of single-pit caveolae at the PM [173]. Therefore, microtubules act as tracks by which caveolae can move from the PM to intracellular locations, in contrast to actin, which contributes to the PM confinement of caveolae [76,173,195]. It has also been described that the protein IQ motif containing GTPase-activating protein 1 (IQGAP1) and integrin-linked kinase (ILK), which are both regulators of cytoskeletal organization, regulate the movement of Cav1 from microtubules to the cortical actin filaments [4,144,196,197].

### 3.3. Intermediate Filaments (IFs) and Caveolae

Less is known about the relationship between IFs and caveolae, but there is an association between Cav1 and different types of IFs, such as keratins [198,199]. Cav1 localizes in keratinocytes with hemidesmosomal integrins, that are coupled with keratin cytoskeleton, and together with Arf6, control its biogenesis and dynamics [200]. The colocalization and association of Cav1 with nestin IFs have also been described [201,202], and it has an impact on transforming growth factor-β (TGF-β) signaling [202]. The most well-studied relationship is between Cav1 and vimentin, which is associated with integrins at focal adhesions. Although there is a cell-dependent regulation of vimentin expression by Cav1 [203,204,205], a clear unidirectional regulation of vimentin on Cav1 has been reported. Vimentin depletion facilitates Cav1 mobilization and increases Y14Cav1 phosphorylation [187,201,205,206]. Finally, regarding IFs type V, which corresponds to the nuclear lamina, the main component of the nucleoskeleton that plays an important role in gene expression regulation and has been described as a link with Cav2. Cav2 can be targeted to the inner nuclear membrane (INM) in response to insulin stimulation and induces Cav2 Tyr19 phosphorylation . It interacts with lamin A/C and acts as an epigenetic regulator of adipogenic genes upon adipogenic stimulation [178,207,208,209].

## 4. Caveolae/Cav1 as Regulators of ECM Composition and Architecture

All tissues and organs have an acellular stromal microenvironment composed of a complex 3D network of proteins, glycoproteins, and polysaccharides called the extracellular matrix (ECM). In the past, the ECM was just thought to play a role in providing structural and mechanical support to preserve tissue integrity and allow cell migration. However, it is now recognized as a physiologically active component of all tissues since it controls a variety of processes such as cell survival, proliferation, or cell fate [210]. The main components of the ECM are elastic fibers, fibrillar collagens, glycosaminoglycans, and proteoglycans. However, under pathological conditions, e.g., tumor progression and metastasis [211], the ECM can exhibit a particular topology [11] and be enriched in specific components, such as tenascin C (TnC) or osteopontin [212,213].

As previously mentioned, ECM-mediated mechanotransduction is an important factor that affects tissue homeostasis [214]. The mechanosensing process involves the assessment of the mechanical properties of the ECM by the cells through specialized structures, such as integrins and caveolae, and through changes in the actomyosin cytoskeleton. In turn, the cells can modify this ECM by two basic mechanisms: (i) by the physical remodeling on ECM applying forces on the matrix and (ii) by the chemical deposition or degradation of different ECM components [215]. Caveolae have been widely studied as mechanosensors in this context, and there is a growing body of evidence supporting the role of caveolae, particularly Cav1, in ECM deposition and remodeling. In fact, Cav1 has been described to have important physiological roles in fibroblasts, the main cell type involved in ECM turnover and reorganization in both homeostatic and pathological conditions on tissue stroma. However, which of the mechanisms discussed below rely on Cav1 alone or are extensive to caveolae remains to be fully elucidated. 

### 4.1. Physical Remodeling of ECM

Cav1 directly promotes the biomechanical remodeling of the ECM through stress-fiber regulation via RhoA [10,11] and YAP [12] (Figure 3).

#### 4.1.1. Rho-Mediated Actomyosin Contraction 

Rho small GTPases play a crucial role in this remodeling of the actin cytoskeleton [181]. Interestingly, these Rho GTPases could also be regulated by caveolae. 

Src is involved in Rac and Cdc42 activation and can inhibit Rho through the activation of p190RhoGAP, an endogenous inhibitor of Rho. Cav1 positively regulates Rho activity in several cell types [10,11,35,188] by affecting the localization and activity of p190RHOGAP, which in turn promotes Rho-dependent stress fiber formation. The phosphorylation of Cav1 at Tyr14 seems to be relevant to these processes through its interaction with Src and Csk [10] Additionally, Cav1 has been linked to the actin cytoskeleton through filamin [154], and a polarized distribution of Cav1 in migrating cells has been observed [10,216,217]. Moreover, caveolar components could also regulate RhoA signaling via other mechanisms, such as the PM targeting of Rac1, physical interactions, and rates of degradation, and influence cytoskeletal regulation [10,28,95,189,190]. In general, although this can vary depending on the cellular context, Cav1 depletion reduces the number of stress fibers, whereas Cav1 overexpression increases the fluorescence intensity of actin and its reorganization [179,180]. All of these mechanisms support the functions of Cav1 in controlling cell motility and polarization, which are additional processes through which Cav1 potentially affects ECM organization. 

Cav1-mediated actin polymerization leads to relevant functional consequences. For example, Goetz et al. [11] found that stromal cells can remodel the ECM by applying forces on the matrix in a Cav1-dependent mechanism. Cav1-deficient cancer-associated fibroblasts (CAFs) show decreased contractility because Cav1 favors cell elongation in 3D cultures and promotes Rho-dependent contraction through the regulation of p190RhoGAP. They observed that Cav1 knockout (KO) fibroblasts generate a softer substrate with a more disorganized collagen matrix and fewer parallel fibronectin (FN) fibers, which hampers cell migration. With these subjacent mechanisms, the authors showed that stromal Cav1 can alter tumor microenvironments, facilitating tumor invasion in vivo. The Tyr14-phosphorylation of Cav1 is presumably relevant to these specific processes through the regulation of Rho and Rac1 activity, since the re-expression of unmodified Cav1, but not its non-phosphorylatable mutant Cav1Y14F, rescued features related to cell morphology and contraction (Rho), as well as to Rac1 localization observed in Cav1KO mouse embryonic fibroblasts (MEFs). 

#### 4.1.2. YAP

One of the main pathways regulated by mechanical stimuli is the Hippo pathway, which is involved in organ size and cell fate regulation [117]. When this pathway is re-pressed by a mechanical input, YAP and TAZ, which are effectors of the Hippo pathway, are translocated into the nucleus, where they act as transcriptional co-regulators through association with TEAD transcription factors [117,218,219].

The resulting biological effects of YAP/TAZ-induced mechanotransduction highly vary across cell types and are also dependent on the nature of the mechanical stimuli—i.e., ECM stiffness and organization, cell density, shear stress, stretching, etc. [116]. YAP has also been shown to regulate ECM in CAFs [220]. In this context, Moreno-Vicente et al. [12] described Cav1 as the upstream regulator of YAP-dependent ECM remodeling, since the impaired ability of Cav1KO MEFs to retract collagen gels or to organize collagen fibers was rescued by the expression of the non-phosphorylatable YAP-5SA version, which in-creases YAP nuclear translocation. This study demonstrated that Cav1 positively regulates YAP activity, modulating the response to ECM stiffness through a mechanism de-pendent on actin cytoskeleton dynamics, but independent of the canonical Hippo path-way. The authors also demonstrated that Rho activity is necessary, but insufficient, for the Cav1-dependent mechanoregulation of YAP.

Soon afterwards, Rausch et al. identified a decreased caveolae density in cells lacking YAP/TAZ [114], highlighting the role of this pathway in controlling the expression of two essential caveolar components, Cav1 and Cavin1, and confirming the reciprocal regulation of the YAP/TAZ axis and caveolae. However, in this study, the authors also observed that the knockdown of Cavin1 and Cav1 decreased the nuclear activity of YAP/TAZ in HEK293A and U2OS cells. Similarly, Cav1 depletion increased the expression of YAP/TAZ target genes in mesothelial cells [115], indicating that Cav1 regulates YAP/TAZ nuclear translocation in a context- and cell-type-dependent manner. 

### 4.2. Chemical Remodeling of ECM

Cav1 can also alter the ECM by modifying its composition, which occurs mainly through the regulation of the TGF-β pathway and exosomal secretion [221]. 

#### 4.2.1. Regulation of TGFB Pathway

The TGFβ pathway promotes the expression of genes involved in the synthesis of collagens and other matrix proteins and also decreases the expression of genes that encode for mediators of ECM degradation, playing a key functional role in the activation of fibro-blasts and other ECM-producing cell types. As previously mentioned, caveolae act as signaling platforms where different membrane receptors can be localized, including TGF-β receptors (TβRs). In 2001, Razani et al. demonstrated the interaction between TβR-I and Cav1 in caveolae, resulting in the inhibition of SMAD (Suppressor of Mothers Against Decapentaplegic) signaling [85]. Later, the localization of the TβR-II receptor to caveolae in endothelial cells has also been demonstrated using density gradient fractionation and co-immunoprecipitation [86]. Caveolae have been shown to promote TβR internalization, thus emerging as an alternative pathway to clathrin-dependent endocytosis of these receptors [90]. However, while the internalization of TβRs via clathrin-coated pits enhances TGF-β signaling, it seems that caveolae-mediated endocytosis promotes TGF-β degradation, thus inhibiting TGF-β signaling [222]. Recently, the glycosylation of tubulin-β2 and tubulin-β3 has been shown to be necessary for caveolae-dependent TGF-β receptor internalization [223].

The relevance of caveolae/Cav1 on TGF-β pathway modulation becomes evident in several studies that account for important functional effects of these processes, resulting in the involvement of Cav1 in several fibrotic diseases [140,142]. For example, Cav1 sup-presses epithelial-to-mesenchymal transition (EMT) and fibrosis during peritoneal dialysis [224]. Moreover, Cav1 deficiency induces mesothelial-to-mesenchymal transition (MMT) through the hyperactivation of the TGF-β1 pathway in response to mechanical stretching: mechanistically, silencing or knocking out Cav1 transcriptionally activates TGF-β1 and TβR-I expression [115] 

#### 4.2.2. Secretion of Exosomes

An additional mechanism through which Cav1 can influence stromal composition is via the exosomal deposition of particular ECM components. Exosomes are extracellular vesicles that originate from the endosomal compartment and are secreted by cells upon the fusion of multivesicular bodies (MVBs) with the plasma membrane. In the last few years, they have gained attention as a key mechanism for intercellular communication [225], since they can deliver a wide variety of cargoes—i.e., proteins, lipids, mRNAs, non-coding RNAs, etc.—to neighbours and even distant cells. 

Although few studies have examined the impact of these vesicles on ECM deposition, in 2020, Albacete-Albacete et al. [221] showed that some ECM components are sorted into exosomes via a Cav1-dependent mechanism. Mechanistically, Cav1, which regulates cholesterol homeostasis and dynamics, modulates the cholesterol content at MVBs, affecting exosomal size, and thus, the sorting of certain cargoes into them. Cav1 deficiency provokes cholesterol accumulation in the endosomal compartment, leading to the production of smaller EVs that are unable to carry high-molecular-weight proteins such as TnC or FN. The authors mimicked the effect of the lack of Cav1 by exogenous cholesterol administration or by the pharmacological inhibition of cholesterol trafficking from the endosomal compartment. However, they also demonstrated that, although exosomal secretion is strictly required for TnC deposition, this is not the case for FN, whose deposition is partially reduced, but not blunted by inhibiting the release of exosomes. However, although Cav1 does not seem to be necessary for FN deposition [221], it has been suggested to participate in FN turnover, apparently through caveolae, by regulating FN internalization and degradation [226]. Moreover, as for collagen fibers, Cav1 has been shown to play a role in FN physical remodeling and fibril orientation [11]. 

As reviewed in [227], TnC favors the establishment of a pre-metastatic niche. Then, the findings of Albacete-Albacete et al. [221], together with those derived from Goetz et al. [11], shed light on the role of Cav1 in promoting cell invasion and metastasis through ECM alteration.

Although the role of Cav1 in exosome-mediated ECM deposition has been discussed, the sorting of ECM components to exosomes may be regulated by additional caveolae constituents, such as Cavin1, which has also been shown to control this process [228].

## 5. Concluding Remarks

Caveolae are key mechanosensor and mechanotransducer structures of the PM that play an important role in buffering tension and initiate and contribute to different mechanotransduction pathways (Figure 4). Even in the absence of proper caveolae, Cav1 in dolines can also provide mechanoadaption. Its connection with the different types of cytoskeletons is especially relevant for this function. Its association with actin fibers is better studied among CSK types, and several models of interplay have been described. However, all of the interactions between caveolae and the cytoskeleton filaments and their physiological impact upon determined mechanical stimuli are not completely understood.

New insights regarding the relationship between caveolae/Cav1 and ECM deposition and remodeling have emerged in the last few decades. Apart from the chemical modification of ECM by caveolae/Cav1-dependent mechanisms, the modulation of actin CSK by caveolae also has a central role ECM reorganization (Figure 4). Caveolae are thus proposed as a central hub where mechanosensing is linked to cellular responses that, in turn, modify the extracellular environment in a reciprocal way. In addition, caveolae and Cav1 can be potentially targeted in a translational context based on their relevance in certain pathological conditions, such as cancer or fibrotic diseases. However, given the cell- and context-dependent effects of Cav1 on cell behavior, further studies are needed to completely envision the utility of caveolae/Cav1-targeting therapies.

## Figures and Tables

**Figure 1 cells-12-00942-f001:**
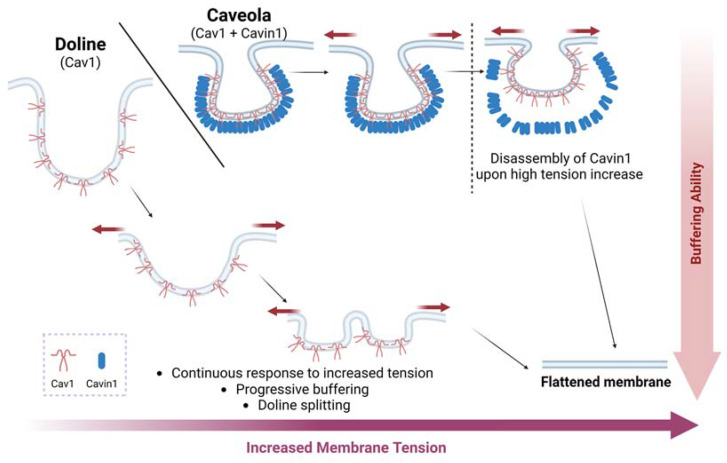
Cav1 can buffer PM tension via different mechanisms. Dolines (bigger and irregular PM invagination devoid of PTRF) gradually buffer low–medium mechanical force increases, while caveolae (regular PM nanoinvaginations, size-restricted by PTRF binding) provide acute buffering to higher forces, only flattening beyond a certain tension threshold. Thus, dolines exhibit a ‘spring-like’ response vs. the ‘mechanical switch’ constituted by caveolae. Created with BioRender.com accessed on 15 January 2023.

**Figure 2 cells-12-00942-f002:**
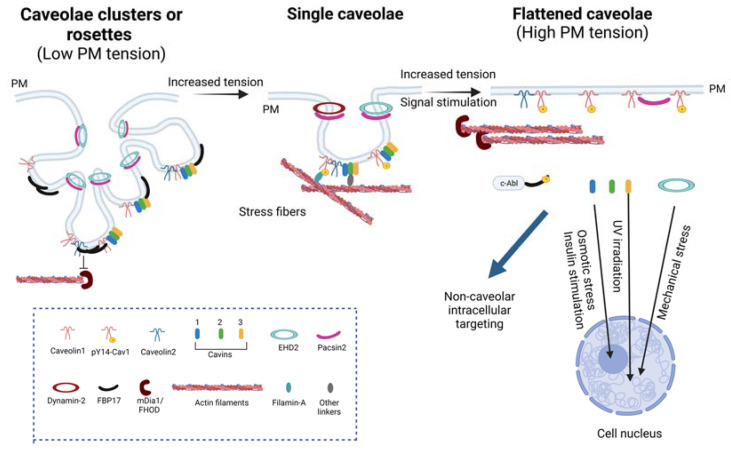
Caveolae composition and changes upon mechanical and signaling challenges. Single caveolae are composed of caveolins, cavins, and accessory proteins, and they may associate with the actin cytoskeleton through Filamin-A and other unknown linkers. Upon mechanical forces or signal stimulation, caveolae can flatten and release some of their components for intracellular targeting, such as Cavin1, Cavin3, and EHD2. This also stimulates FBP17 phosphorylation by c-Abl and its dissociation. When PM tension is reduced, caveolae organize into rosettes that require FBP17, which inhibits formin mDia. Furthermore, Cav1 could be phosphorylated at Y14 in response to mechanical stimuli. Created with BioRender.com accessed on 1 March 2023.

**Figure 3 cells-12-00942-f003:**
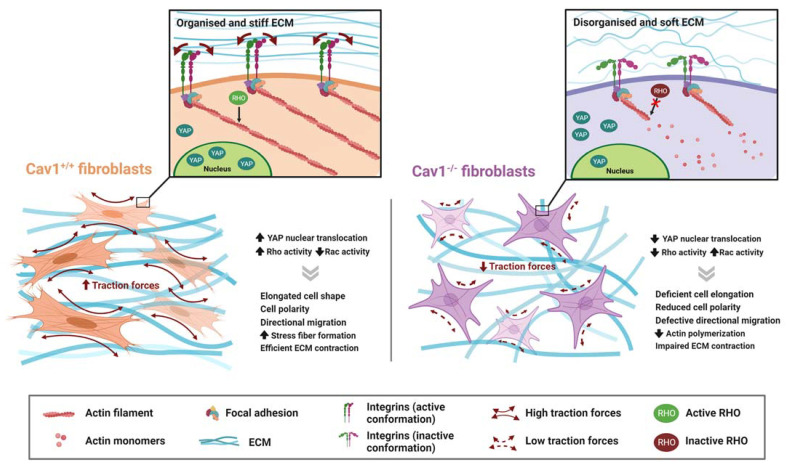
Role of Cav1 in ECM physical remodeling. Stromal cells interact with the ECM, reorganizing it by applying forces at cell–ECM contact sites. In fibroblast, Cav1 promotes actin polymerization and stress fiber formation through Rho activation, and regulates cell polarization and directional migration. In response to substrate stiffness, Cav1 also positively regulates YAP nuclear translocation, which in turn regulates the expression of genes involved in ECM remodeling. Cav1 favors ECM contraction through the exertion of tractional forces by fibroblasts, leading to a more organized and stiffer ECM, while Cav1-depleted fibroblasts generate a disorganized and softer ECM. Created with BioRender.com accessed on 6 March 2023.

**Figure 4 cells-12-00942-f004:**
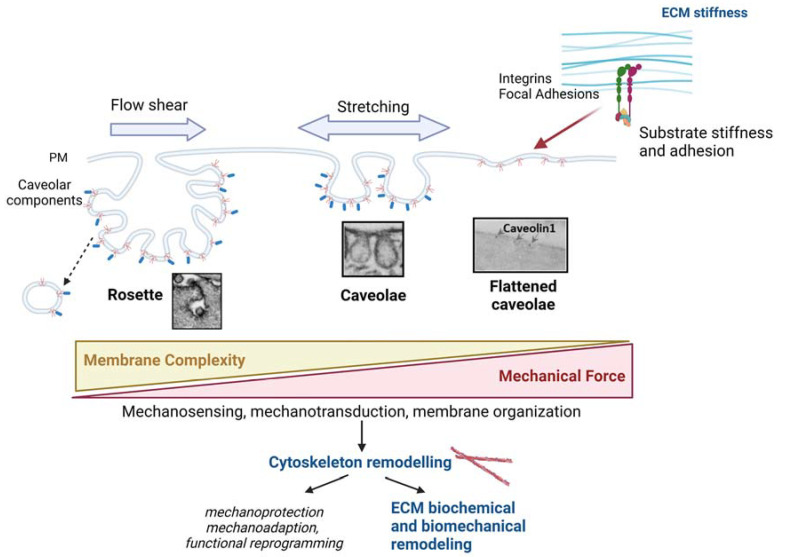
Schematic integrating the role of caveolae in mechanosensing (including ECM stiffness) and mechanotransduction (including cytoskeleton reshaping, and subsequent ECM remodelling). Caveolar structures exhibit high plasticity, such that upon reduced PM tension higher-order structures named ‘rosettes’ -vacuoles surrounded by clusters of caveolae connected through a single neck to the PM- are formed. Rosettes can flatten in response to mechanical challenges, such as shear stress, stretching or substrate stiffness, leading to the formation of single-pit caveolae. Further PM tension increase results in complete flattening and thus caveolae disassembly. Therefore, there is an inverse correlation between the complexity of Cav1-bound membrane and PM tension. Regulatory pathways integrate to control the plasticity, trafficking and mechanosensing and mechanostransducing abilities of caveolae, which lead to changes in the cytoskeleton, which in turn results in ECM remodeling through different effector mechanisms and in different functional outcomes for adapting to the surrounding mechanical stimuli. Created with BioRender.com accessed on 6 March 2023.

## Data Availability

No new data were created.

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
