# Peer review of "Caveolae Mechanotransduction at the Interface between Cytoskeleton and Extracellular Matrix"

_cells, 2023, doi:10.3390/cells12060942_

Round 1

Reviewer 1 Report

This timely manuscript reviews the role of caveolae as an important player in mechanotransduction. This review covers up-to-date findings on the fairly comprehensive topics including its composition and organization, its role in diseases, its interaction with other intracellular and extracellular components. This manuscript is well organized, but some sub-titles do not seem to be necessary and can be regrouped or renamed. Also, inclusion of a summary schematic will help readers. Therefore, I recommend a manuscript for a very minor revise and resubmisison.

1. A schematic summarizing the manuscript, (i.e., the role of caveolae and its interaction with the cytoskeleton and ECM) will greatly help readers.

2. It sounds more logical to discuss the organization of caeolae and then discuss its structure. Thus, the sections 2.2 and 2.1 can be switched. Also, these two sections could be combined as a single section, such as “2.1 Caveolae organization and composition”.

 3. The title of the section 3 is vague. More specific title would help readers, such as “Interaction between caveolae and cytoskeleton”.

4. On line 422, the section 4.2 should read “4.1”

Author Response

This timely manuscript reviews the role of caveolae as an important player in mechanotransduction. This review covers up-to-date findings on the fairly comprehensive topics including its composition and organization, its role in diseases, its interaction with other intracellular and extracellular components. This manuscript is well organized, but some sub-titles do not seem to be necessary and can be regrouped or renamed. Also, inclusion of a summary schematic will help readers. Therefore, I recommend a manuscript for a very minor revise and resubmisison.

We appreciate the positive and constructive comments of this reviewer.

  1. A schematic summarizing the manuscript, (i.e., the role of caveolae and its interaction with the cytoskeleton and ECM) will greatly help readers.

    As the review covers very broad topics like cytoskeleton and ECM, it was difficult to summarize the relevance of caveolae with these two components in a single scheme. Figures 2 and 3 together summarize the role of caveolae on each of these components separately, which we think could be easier for understanding instead of a very complex scheme. However, in light of this reviewer´s view we have generated a simple schematic (Fig. R1, for Reviewer 1`s inspection only at this point) merging the two functions regulating/regulated by caveolae. In summary, it shows that among the mechanical forces sensed by caveolar structures is ECM (i.e. substrate) stiffness, which after caveolae-mediated transduction, it leads to cytoskeleton remodelling, which impacts back on ECM remodelling. If the reviewer and editors find this scheme appealing and necessary to help readers, we can include it in the revised version of the manuscript. Alternatively, we can also refer to Figure 4 in our previous review (Parton and del Pozo, Nat. Rev. Mol Cell Biol 2013) showing the same; the problem of that figure is that ECM stiffness is not included as example of mechanical force detected and transduced by caveolae, but it shows that caveolae regulate Rho-driven acto-myosin cytoskeleton contraction, which in turn drives ECM remodelling (which can in turn be sensed by caveole).                 

    Figure R1

Figure R2

2. It sounds more logical to discuss the organization of caveolae and then discuss its structure. Thus, the sections 2.2 and 2.1 can be switched. Also, these two sections could be combined as a single section, such as “2.1 Caveolae organization and composition”.

We appreciate the comment and combined the two sections into a single one called “2.1. Caveolae composition and organization” as proposed. 

3. The title of the section 3 is vague. More specific title would help readers, such as “Interaction between caveolae and cytoskeleton”.

We appreciate the feedback and we have changed the subtitle to the recommended one “3. Interaction between caveolae and cytoskeleton”

4. On line 422, the section 4.2 should read “4.1”

We corrected the erratum

Reviewer 2 Report

Comments on the review “Caveolae mechanotransduction at the interface between cyto-2 skeleton and the extracellular matrix” by Laura Sotodosos-Alonso et al.

General comments:

line 50: in order to be consistent with the other titles, 2. needs to be completed, for instance: “Caveolae components and organization”.

line 90: I'm not sure that "in our laboratory" is necessary to specify in the context of a review!

line 118 to 126: enumeration of a sequence of steps, difficult to follow. I suggest to specify from the beginning that "(l 125-126) lipids play a crucial role in caveolae formation and stabilization and the recruitment of most caveolar components", thus the following explanation (line 118 to 124) would be easier to read and understand.

line 179: I have a naive question: since caveolae buffer membrane tension, what about their effect on mechanosensitive channels activity? Do they modify, for instance, the threshold activation of the channel. Is there any information on the relationship between caveolae and these important player (MS-channel) of the mechanotransduction?

line 214: YAP, TAZ, etc, when they are used for the first time in the text, the meaning of abbreviation/acronym should be explained.  

same for SMAD line 507.

Figure 2: The legend should mention what represent the scheme in the bottom right corner of the figure; a cell, a nucleus?

Figure 3: In order to be more understandable the figure should be improved. How are tensile forces generated? A zoom on the contact points between the cell and ECM should allow to understand the forces applied to the system. What do the arrows pointing upwards and those pointing downwards mean in the schemes and in the columns on the right?

Minor comments:

line 422: 4.1 instead of 4.2

line 528: remove “t”

Figure 2: colors of Caveolin 1 and Caveolin 2 are not distinguishable (at least by color blind people).

Conclusion:

This review brings a lot of information on the structure and role of caveolae in non-walled cells.

As a non-expert in caveolae (my field is the biomechanics of the membrane), I learned a lot. On the other hand, to make the manuscript understandable by a wider public it would be necessary (as indicated above) to improve the figures (more details better caption). Also, to help the reader, a list of abbreviation would be welcome.  

Author Response

Comments on the review “Caveolae mechanotransduction at the interface between cyto-2 skeleton and the extracellular matrix” by Laura Sotodosos-Alonso et al.

General comments:

  • line 50: in order to be consistent with the other titles, 2. needs to be completed, for instance: “Caveolae components and organization”.

We appreciate the comment and we have thus changed the title to “2. Caveolae: composition, organization and function” to include the three main topics that are discussed in this Section.

  • line 90: I'm not sure that "in our laboratory" is necessary to specify in the context of a review!

We apologize for this lapsus, appreciate the feedback and consequently have eliminated this sentence.

  • line 118 to 126: enumeration of a sequence of steps, difficult to follow. I suggest to specify from the beginning that "(l 125-126) lipids play a crucial role in caveolae formation and stabilization and the recruitment of most caveolar components", thus the following explanation (line 118 to 124) would be easier to read and understand.

We appreciate the comment and accordingly have changed the order to make it easier to understand as: “Lipids are crucial in caveolae formation and stabilization and the recruitment of most caveolar components [13,73].  Alterations in cholesterol content also significantly affect Cav1 and caveolae, as cholesterol depletion by methyl-β-cyclodextrin treatment leads to the dissociation of cavins, and caveolae flatten [5,71] Moreover, cavins bind to PtdIns(4,5)P2 (phosphatidylinositol 4,5-biphosphate) and phosphatidylserine (PtdSer) [43,66]. EHD2 can bind to PtdIns(4,5)P2 [72], requires cholesterol to bind to caveolae [54], and neck morphology is subjected to changes in lipid accumulation [73]. Pacsin2 and FBP17 can also bind phosphatidylinositol [74]. “

  • line 179: I have a naive question: since caveolae buffer membrane tension, what about their effect on mechanosensitive channels activity? Do they modify, for instance, the threshold activation of the channel. Is there any information on the relationship between caveolae and these important player (MS-channel) of the ?

This a very good question that have intrigued us very much as well during the years, to the point of pursuing this idea in an incipient project. Not much is known, apart from the fact that many mechanosensitive channels including the mechanosensitive “Transient receptor potential” (mTRP or TRPM or TRPV families…) channels, the Piezo family, among others, localize in caveolae. We could speculate that caveolae flattening and subsequent lipid bilayer remodelling would affect the conformation, 3D structure and/or localization of these channels and hence affect their function, but this is actually ongoing research in our lab and probably other labs.

  • line 214: YAP, TAZ, etc, when they are used for the first time in the text, the meaning of abbreviation/acronym should be explained

We apologize for this omission, and now haveincluded the meaning of these proteins when they are used for the first time: “YAP (yes-associated protein) and TAZ (Transcriptional coactivator with PDZ-binding motif)

  • same for SMAD line 507

We included the meaning of the abbreviation: “SMAD (Suppresor of Mothers Against Decapentaplegic)”

  • Figure 2: The legend should mention what represent the scheme in the bottom right corner of the figure; a cell, a nucleus?

We appreciate the feedback and realized that the label of the cell nucleus is missing, so we included it in the figure 2.

  • Figure 3: In order to be more understandable the figure should be improved. How are tensile forces generated? A zoom on the contact points between the cell and ECM should allow to understand the forces applied to the system. What do the arrows pointing upwards and those pointing downwards mean in the schemes and in the columns on the right?

We appreciate the feedback and we have improved Figure 3 accordingly. We expect that, with these modifications, the reader can better understand how the mechanical forces exerted by both Cav1+/+ and Cav1 -/- fibroblasts are able to remodel or not, respectively, the ECM.

Minor comments:

  • line 422: 4.1 instead of 4.2

We corrected the erratum

  • line 528: remove “t”

We corrected the erratum

  • Figure 2: colors of Caveolin 1 and Caveolin 2 are not distinguishable (at least by color blind people).

We appreciate the feedback and have changed Caveolin 2 to a darker colour to make it more easily distinguishable from Caveolin1’s colour.

Conclusion:

This review brings a lot of information on the structure and role of caveolae in non-walled cells.

As a non-expert in caveolae (my field is the biomechanics of the membrane), I learned a lot. On the other hand, to make the manuscript understandable by a wider public it would be necessary (as indicated above) to improve the figures (more details better caption). Also, to help the reader, a list of abbreviation would be welcome. 

We appreciate the positive and constructive comments of this reviewer and are glad s/he finds it interesting and in/formative. Regarding abbreviations, there are not so many (most of them are already included here like Cav1, PTRF, YAP/TAZ, SMAD… but are happy to include such a table if the format permits it (in the revised version of the text, next step according to the editorial office).
